# Heterogeneous multimeric structure of isocitrate lyase in complex with succinate and itaconate provides novel insights into its inhibitory mechanism

**Sunghark Kwon[1]ᵒ, Hye Lin Chun[2]ᵒ, Hyun Ji Ha[2], So Yeon Lee[2], Hyun Ho Park[2]***

**1** Department of Biotechnology, Konkuk University, Chungju, Chungbuk, Republic of Korea, **2** College of Pharmacy, Chung-Ang University, Dongjak-gu, Seoul, Republic of Korea

ᵒ These authors contributed equally to this work.
* xrayleox@cau.ac.kr

**Data Availability Statement:** The coordinates and structure factors of the final model have been deposited in the Protein Data Bank (PDB). The PDB access code is 7CP1.

## Abstract

During the glyoxylate cycle, isocitrate lyases (ICLs) catalyze the lysis of isocitrate to glyoxylate and succinate. Itaconate has been reported to inhibit an ICL from *Mycobacterium tuberculosis* (tbICL). To elucidate the molecular mechanism of ICL inhibition, we determined the crystal structure of tbICL in complex with itaconate. Unexpectedly, succinate and itaconate were found to bind to the respective active sites in the dimeric form of tbICL. Our structure revealed the active site architecture as an open form, although the substrate and inhibitor were bound to the active sites. Our findings provide novel insights into the conformation of tbICL upon its binding to a substrate or inhibitor, along with molecular details of the inhibitory mechanism of itaconate.

## Introduction

Isocitrate lyases (ICLs; isoforms 1 and 2) catalyze the reversible conversion of isocitrate to glyoxylate and succinate in the glyoxylate cycle, which is used to bypass the two decarboxylation steps of the tricarboxylic acid (TCA) cycle (Fig 1A) [1, 2]. This catalytic reaction can be reversed to produce isocitrate from glyoxylate and succinate [1, 2]. The ICL-initiated glyoxylate cycle is essential for various organisms, including bacteria, fungi, and plants, to reactivate the TCA cycle in circumstances that limit carbon supply as an energy source by offering alternative energy sources such as fatty acids [3, 4]. ICLs from *Mycobacterium tuberculosis*, a bacterium causing tuberculosis, can also catalyze the lysis of 2-methylisocitrate to produce pyruvate and succinate [5].

The upregulation of the glyoxylate cycle via ICLs is necessary for the survival of fungi and bacteria in the host after infection [3]. *M. tuberculosis* also harbors ICLs that are involved in its persistence and virulence [6]. Owing to the crucial function of ICLs for the survival of *M. tuberculosis* in the host, they have been considered major therapeutic targets for the treatment of tuberculosis [7, 8]. Several inhibitors of ICLs—natural and synthetic products—have been reported [9, 10], including bromopyruvate, nitropropionate, and itaconate; however, these

**Funding:** This research was supported by the National Research Foundation (NRF) of the Ministry of Education, Science, and Technology (NRF-2017M3A9D8062960, NRF-2021R1A2C3003331, NRF-2018R1A4A1023822, and NRF-2020R1G1A1100765).

**Competing interests:** The authors have declared that no competing interests exist.

products, unfortunately, inhibit host metabolic enzymes other than ICLs via non-specific binding [10]. Hence, there remains a need to identify therapeutic agents that can selectively inhibit ICLs, which will be ideal for the treatment of tuberculosis.

Itaconate, an unsaturated dicarboxylic acid, is an intermediate metabolite in the TCA cycle and is produced by the decarboxylation of *cis*-aconitate [11]. This process is catalyzed by *cis*-aconitate decarboxylase (encoded by immune-responsive gene 1) [12, 13]. Macrophage lineage cells and various fungi are known to produce itaconate during infection in a bid to remove the pathogen [14–16]. The antibacterial activity of itaconate is accomplished by direct inhibition of glyoxylate cycle-related enzymes, such as ICLs [13, 17]. A recent study reported that mammalian cells that produce itaconate after viral infection inhibit the activity of succinate dehydrogenase, thereby inhibiting the replication of Zika viruses in infected neurons [18]. Although it has been shown that itaconate produced by a host directly inhibits an ICL from *M. tuberculosis* (tbICL) to protect against the infection [13, 19], the molecular mechanism of ICL inhibition by itaconate remains elusive.

There are two tbICL isoforms (tbICL1 and tbICL2) [6, 20], which are structurally very different but play important roles in carbon catabolism in *M. tuberculosis* [6, 20]. Further, tbICL1

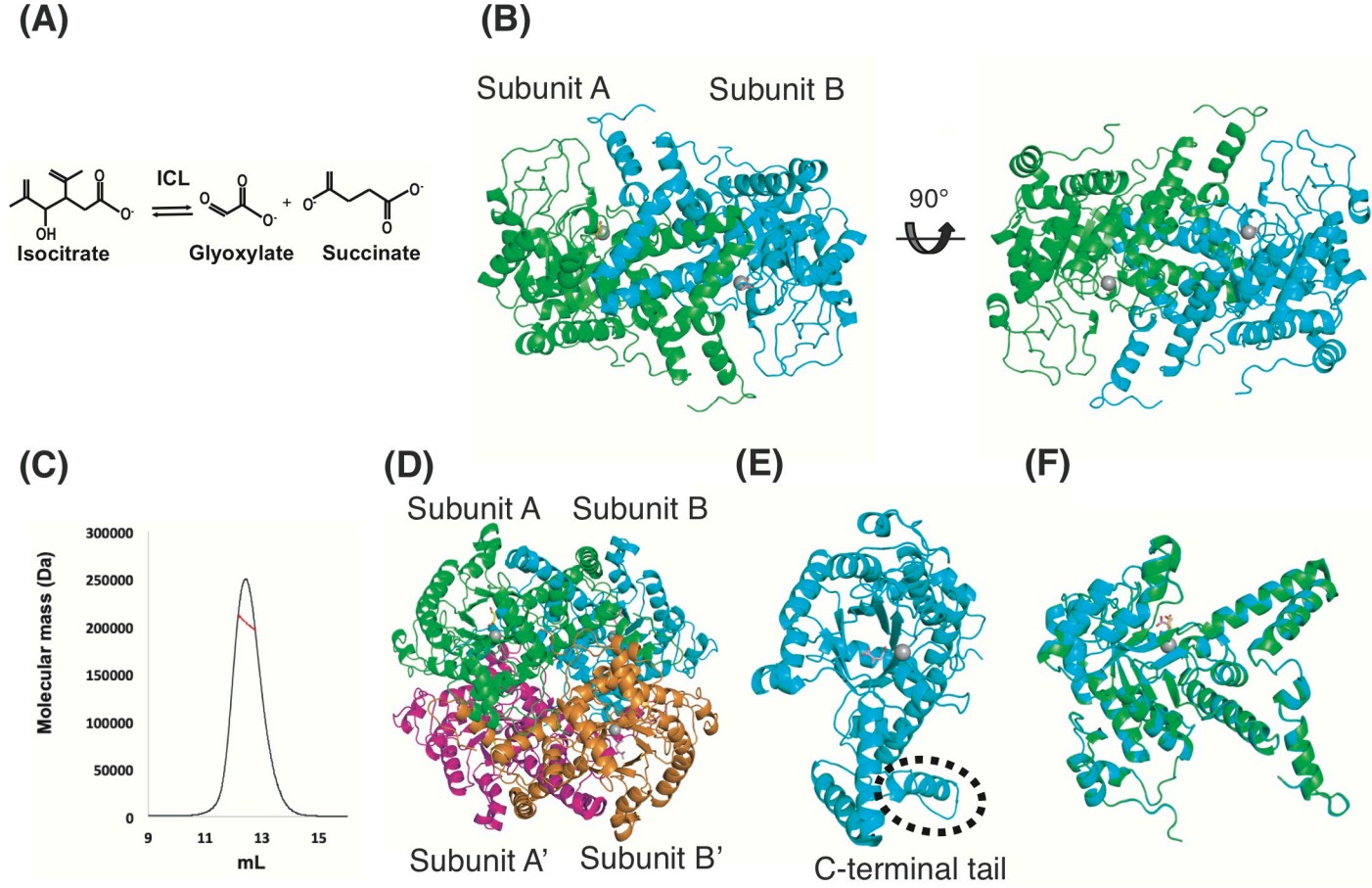

**Fig 1. Overall structure of tbICL in complex with succinate and itaconate.** (A) Reversible catalytic reaction of tbICL, involving isocitrate, glyoxylate, and succinate. (B) Crystal structure of tbICL in the asymmetric unit. The two molecules (subunits A and B) are represented graphically from two different viewpoints. The gray spheres, yellow stick (subunit A), and pink stick indicate $Mg^{2+}$ ions, succinate, and itaconate, respectively. (C) SEC-MALS profile of tbICl. Measured values using SEC-MALS (red) are plotted as SEC elution volume (x-axis) versus absolute molecular mass (y-axis) distributions on the SEC chromatogram (black) at 280 nm. (D) Tetrameric structure of tbICL. Two 2fold symmetric neighboring molecules (magenta and orange) were identified and modeled with the dimeric structure in the asymmetric unit. (E) The monomeric structure of tbICl (subunit B). (F) Superimposition of subunit B onto subunit A of tbICL.

undergoes major conformational changes of some specific regions to close the active site upon binding its inhibitors, such as 3-bromopyruvate and 3-nitropropionate [21], whereas tbICL2 is activated upon binding acetyl-CoA and propionyl-CoA [20]. A recent study also proposed two possible catalytic pathways of tbICL1 using quantum mechanics/molecular mechanics [22]. However, structural information on other inhibitors and their working mechanisms of tbICL is still insufficient.

To elucidate the molecular mechanism of inhibitory activity of itaconate on tbICL and to obtain structural insight into the conformational changes upon its binding to its substrates or inhibitors, we determined the structure of tbICL1 in complex with itaconate. In our structure, succinate (a substrate/product of tbICL) and itaconate (an inhibitor of tbICL) were bound to the active sites of chains A and B, respectively. The heterogeneous multimeric structure showed an unexpected conformation of regions proximal to the active site in response to binding of the substrate and inhibitor. Our structural analysis in the present study provides novel insights into determinants for the conformational changes near the active site and into developing new antibiotics targeting tbICL.

## Methods

### Cloning, overexpression, and purification

The gene encoding tbICL was synthesized by Bionics (Daejeon, Republic of Korea) and inserted into a pET21a plasmid vector (Novagen, WI, USA). *Nde*I and *Xho*I restriction sites were used for constructing a recombinant plasmid. The resulting recombinant vector, containing the full-length gene (encoding residues 1–428), was delivered into *Escherichia coli* strain BL21(DE3) competent cells. The cells were cultured at 37˚C in 1 L lysogeny broth containing 50 μg/mL kanamycin. When the optical density value at 600 nm reached 0.75, the temperature was adjusted to 20˚C and 0.5 mM isopropyl β-ᴅ-1-thiogalactopyranoside was added to induce gene expression. The cells were cultured for 18 h in a shaking incubator. Cultured cells were harvested and resuspended in a lysis buffer (containing 20 mM Tris-HCl (pH 8.0), 500 mM sodium chloride, and 25 mM imidazole) and lysed by ultrasonication at 4˚C. The cell lysate and supernatant were separated by centrifugation at 10,000 *g* for 30 min at 4˚C. The collected supernatant was mixed with Ni-nitrilotriacetic acid affinity resin for 2 h, and the mixture was loaded onto a gravity-flow column (Bio-Rad, Hercules, CA, USA). The resin was washed with 50 mL washing buffer (containing 20 mM Tris-HCl (pH 8.0), 500 mM NaCl, and 60 mM imidazole), and the resin-bound target protein was eluted from the resin using elution buffer (containing 20 mM Tris-HCl (pH 8.0), 500 mM NaCl, and 250 mM imidazole). The eluted protein was pooled, concentrated, and injected in a Superdex 200 10/300 GL column (GE Healthcare, Waukesha, WI, USA) on the ÄKTA Explorer system (GE Healthcare, Chicago, IL, USA), which had been pre-equilibrated with a solution containing 20 mM Tris-HCl (pH 8.0) and 150 mM NaCl. The eluted protein was concentrated to 7.8 mg/mL for further utilization. Purity of the protein was visualized and assessed using sodium dodecyl sulfate–polyacrylamide gel electrophoresis.

### Crystallization and X-ray diffraction data collection

The tbICL protein was crystallized using the hanging drop vapor diffusion method at 20˚C. Commercial kits, such as Crystal Screen, Crystal Screen 2, Index, Natrix (Hampton Research), and Wizard Classic I and II (Rigaku Reagents), were used for crystallization screening. Initial crystals were obtained by equilibrating a mixture containing 1 μL protein solution (20 mM Tris-HCl (pH 8.0), and 150 mM NaCl) and 1 μL reservoir solution (0.4 M sodium phosphate/ 1.6 M potassium phosphate, 0.1 M imidazole (pH 8.2), and 0.2 M NaCl) against 0.3 mL

reservoir solution. The conditions for crystallization were further optimized by varying protein and precipitant concentrations and buffer pH values. The best crystals were obtained from a crystallization buffer containing 0.6 M sodium phosphate/1.2 M potassium phosphate, 0.1 M imidazole (pH 8.5), and 250 mM NaCl, over approximately 3 days. A single well-formed crystal was selected and soaked in the reservoir solution supplemented with 40% (v/v) glycerol for cryo-protection. To obtain crystals containing substrates and itaconate, tbICL crystals were soaked in crystallization buffers supplemented with respective compounds (each at 5 mM) in various combinations: (1) itaconate, (2) isocitrate, (3) succinate, (4) glyoxylate, (5) succinate + glyoxylate, (6) isocitrate + itaconate, (7) succinate + itaconate, (8) glyoxylate + itaconate, and (9) succinate + glyoxylate + itaconate. X-ray diffraction data were collected at −178˚C on beamline BL-5C at Pohang Accelerator Laboratory (Pohang, Korea). Indexing, integration, and scaling were conducted using HKL2000 [23].

### Structure determination and refinement

tbICL structure was determined by the molecular replacement method, using Phaser [24]. The previously reported tbICL structure (PDB ID: 1F8M) [21] was used as a search model. The initial model was built automatically using AutoBuild in Phenix, and further model building and refinement were performed using Coot [25] and phenix.refine in Phenix [26], respectively. The stereochemical quality of the model was validated using MolProbity [27]. All structural figures were generated using PyMOL [28] and LigPlot$^+$ [29].

### Multi-angle light scattering analysis

The absolute molecular weight of tbICL in solution was measured using SEC-coupled multi-angle light scattering (SEC-MALS). The protein sample was loaded onto a Superdex 200 Increase 10/300 GL (24 mL) column pre-equilibrated with SEC buffer (containing 20 mM Tris-HCl (pH 8.0) and 150 mM NaCl). The flow rate of the buffer and measurement temperature were maintained at 0.4 mL/min and 20˚C, respectively. A DAWN-TREOS MALS detector was systemically connected to an ÄKTA Explorer system. The molecular weight of bovine serum albumin was measured and used as a reference value. All data were processed and assessed using the ASTRA program.

## Results and discussion

### Overall structure of tbICL in complex with succinate and itaconate

Among the crystals prepared using various combinations of substrates and itaconate, the structure of tbICL in complex with succinate and itaconate was determined at 2.58 Å resolution. X-ray diffraction data and refinement statistics for tbICL are summarized in Table 1. The crystal belongs to space group $P6_522$, with two molecules (subunits A and B) present in the asymmetric unit (Fig 1B). It has been assumed that ICLs function biologically as tetramers, although they are stable in solution as dimers [21]. To confirm the multimeric state of tbICL in solution, we performed SEC-MALS, which can provide information on the absolute molecular mass in solution. The molecular mass of tbICL as determined by this method was 203.9 kDa (0.42% fitting error; Fig 1C). Considering 47.08 kDa as the theoretical molecular mass of tbICL (as a monomer), including the C-terminal His-tag, the measured value indicates that tbICL exists dominantly in tetrameric form in solution. Because the oligomerization of proteins tends to depend on several factors, such as protein concentration, temperature, and salt [30, 31], we performed SEC-MALS under various conditions and obtained the same results. Based on previous study findings and our SEC-MALS result, we conclude that tbICL exists in

**Table 1. Data collection and refinement statistics for tbICL in complex with succinate and itaconate.**

| Data collection | tbICL in complex with succinate and itaconate |
|---|---|
| X-ray source | BL 5C beamline |
| Wavelength (Å) | 1.0000 |
| Space group | $P6_522$ |
| Unit cell parameter: | |
| $a, b, c$ (Å) | $a = b = 131.32$, $c = 284.88$ |
| $\alpha, \beta, \gamma$ (°) | $\alpha = \beta = 90$, $\gamma = 120$ |
| Total reflections | 462673 |
| Unique reflections[1] | 44429 (2800) |
| Multiplicity[1] | 10.4 (9.4) |
| Completeness (%)[1] | 95.1 (92.4) |
| Mean $I/\sigma(I)$[1] | 9.0 (3.0) |
| $R_{merge}$ (%)[1,2] | 15.1 (67.0) |
| Resolution range (Å)[1] | 50.00–2.58 (2.64–2.58) |
| **Refinement** | |
| Resolution range (Å) | 43.04–2.58 (2.67–2.58) |
| No. of reflections of working set | 43071 |
| No. of reflections of test set | 2119 |
| $R_{work}$ (%) | 19.4 |
| $R_{free}$ (%) | 15.4 |
| Ramachandran plot: | |
| Favored/outliers (%) | 98.07/0.00 |
| Rotamer outliers (%) | 0.00 |
| RMSD bonds (Å)/angles (°) | 0.008/1.006 |

[1] Values for the outermost resolution shell in parentheses

[2] $R_{merge} = \Sigma_h \Sigma_i |I(h)_i - <I(h)>|/\Sigma_h \Sigma_i I(h)_i$, where $I(h)$ is the observed intensity of reflection h and $<I(h)>$ is the average intensity obtained from multiple measurements.

tetrameric form in solution. We searched for two other appropriate subunits to satisfy the tetrameric state and found two additional symmetric subunits. This other dimeric molecule, with a twofold symmetry, was well positioned with the aforementioned dimeric structure, forming a tetramer (Fig 1D).

The monomeric structure of tbICL consists of sixteen α-helices (α1–16), five $3_{10}$-helices (η1–5), twelve β-strands, and several loops (L1-6). The central eight β-strands form a β-barrel, in which all the strands run parallel to each other (Fig 1E). Our overall structure is nearly identical to the ones reported previously [21]. The active site, containing an $Mg^{2+}$ ion, which is considered to be derived from LB medium, is positioned on the top of the barrel (as discussed in detail in the next section). It is noteworthy that the C-terminal tail from one subunit is known to play an important role in closing the active site of the other subunit [21]. The C-terminal tail contains an α-helix (Fig 1E). In the present study, although the full-length gene (encoding residues 1–428) was expressed, 10 residues at the C terminus could not be built, owing to poor electron density, which was untraceable.

In our structure, subunits A and B contain succinate and itaconate, respectively, in their active sites (Fig 1B; as discussed in detail in the next section). Crystals obtained from the crystallization buffer were soaked in the same solution supplemented with succinate and itaconate to prepare crystals in complex with succinate and itaconate. It is possible that incorporating different compounds into the respective active sites of the two subunits gives rise to

conformational differences. Accordingly, to investigate structural differences, the two overall structures were superimposed (Fig 1F). Structural comparison analysis showed that the root-mean-square deviation (RMSD) value over 416 Cα atoms was 0.24 Å. This value signifies that the two subunits are almost identical to each other structurally. Thus, although each subunit contains a different molecule in the active site, this distribution of heterogeneous molecules in the two subunits did not cause significant structural differences.

## Active site structure of tbICL in complex with succinate and itaconate

The active sites of tbICL are located near the interface of the two subunits (Fig 2A). The entrance to the active site is also positioned near the interface. As mentioned previously, the C-terminal tail from one subunit serves as a lid in closing the active site on the other subunit after substrate entry. In this respect, a dimeric structure is the least functional unit necessary for completing the catalytic reaction of tbICL. In addition, each subunit has a relatively deep cavity as its active site, and an $Mg^{2+}$ ion is positioned at the bottom of the cavity (Fig 2A).

Analysis of the surface electrostatic potential of tbICL revealed the distribution of charged residues. Remarkably, we found that negatively charged residues are dominantly distributed in the active site pocket and that positively charged areas are positioned near the active site (Fig 2B). Such spatial distributions seem to be optimized for accepting positively charged species, such as an $Mg^{2+}$ ion. This observation suggests that the electric field generated in the proximity of the active site plays a crucial role in attracting an $Mg^{2+}$ ion into the active site. However, considering isocitrate as a substrate and glyoxylate and succinate as products, each having a carboxylic acid group, it does not seem logical that the active site would accept those negatively charged molecules. It is thermodynamically unfavorable for a negatively charged substrate to

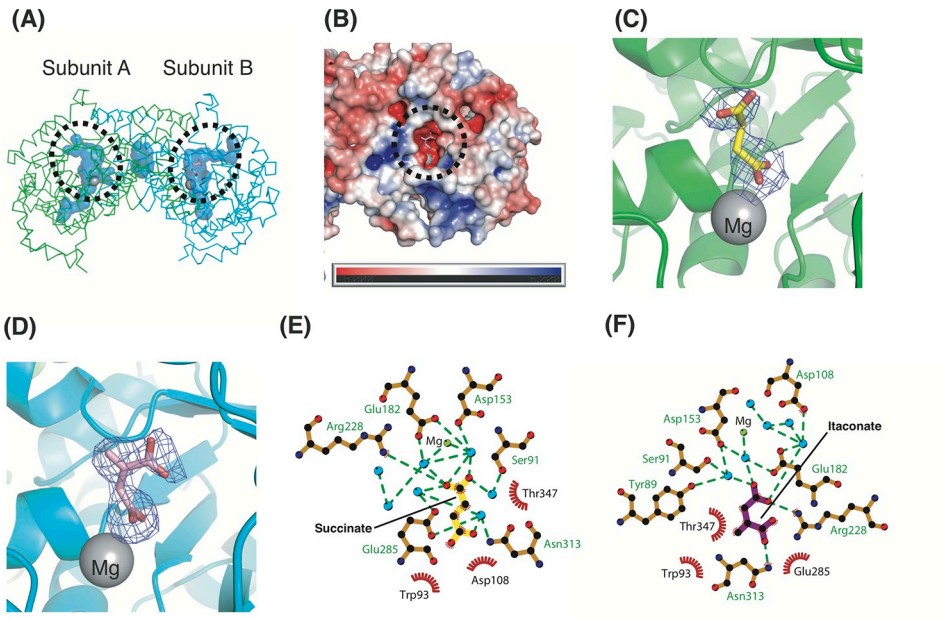

**Fig 2. Active site structure of tbICL in complex with succinate and itaconate.** (A) Cavities including active sites are colored blue. The dashed ovals indicate active sites. (B) Surface electrostatic potential of tbICL. The scale bar ranges from −5 kT/e (red) to 5 kT/e (blue). The dashed circle indicates the active site in subunit B. Omit maps of succinate (C) and itaconate (D). The omit maps ($F_O − F_C$) of succinate and itaconate are colored blue and contoured at the 2.0 sigma and 3.0 sigma level, respectively. Diagrams of succinate (E) and itaconate (F) interactions in the active sites. They interact with water molecules (blue circles) and neighboring residues.

bind to a negatively charged active site. Accordingly, the presence of charged residues in the active site may be necessary to accept an $Mg^{2+}$ ion rather than the substrate. In this respect, it is assumed that the active site recognizes the substrate on the basis of its specific shape, accepting some thermodynamic disadvantages. This finding implies that positively charged compounds with a shape suitable for the active site are possibly potent inhibitors for tbICL.

We found that succinate and itaconate bind to the active sites of subunits A (Fig 2C) and B (Fig 2D), respectively. Their omit maps show that they are well matched to our models (Fig 2C and 2D). They are positioned in the vicinity of the $Mg^{2+}$ ion at the active site. Notably, our structure constitutes a heterogeneous dimer in the asymmetric unit, despite the structure obtained from the same crystal. This finding implies that succinate and itaconate bind competitively to the active site to some degree.

The heterogeneous dimeric structure shows that succinate and itaconate interact with neighboring residues and water molecules in the active site. In the active site of subunit A, two carboxylic acid groups of succinate are coordinated to adjacent water molecules, by which interactions with residues such as Ser91, Trp93, Asp108, Asp153, Glu182, Arg228, Glu285, Asn313, and Thr347 are mediated (Fig 2E). Two water molecules are also coordinated with the $Mg^{2+}$ ion (Fig 2E). In addition, it was observed that the four carbon atoms of succinate form hydrophobic interactions with adjacent residues, such as Trp93, Asp108, and Thr347. Consequently, electrostatic interactions, including the hydrogen-bond network and hydrophobic interactions, contribute to the binding of succinate to the active site. In the active site of subunit B, the same residues that are involved in the succinate binding are associated with interactions with itaconate, except for Tyr89 (Fig 2F). As in the coordination of succinate, itaconate also forms many bonds mediated by water molecules (Fig 2F). However, itaconate directly interacts electrostatically with proximal residues, such as Arg228 and Asn313, whereas succinate does so only with Arg228. In addition, while succinate directly binds to five water molecules, itaconate directly interacts with only three water molecules. These differences suggest that itaconate is relatively less dependent on water molecules for binding to the active site than succinate is.

## Structural comparison with other conformers

A previous study reported that tbICL exists in two forms: open and closed [21]. In the open form, a loop near the active site and the C-terminal tail is located slightly away from the active site, thereby facilitating access of substrate or water molecules to the active site. Conversely, in the closed form, the two regions (the loop and the C-terminal tail) play a mechanical role in closing the active site by moving toward it. The study also showed that inhibitors such as 3-bromopyruvate and 3-nitropropionate induce conformational changes from the open form to the closed form [21]. Based on the structures in complex with the inhibitors, which structurally mimic the substrate moiety, it has been assumed that substrate binding induces the closure of the active site by moving the loop and the C-terminal tail toward the site.

In this study, we present a heterogeneous dimeric structure simultaneously containing succinate (substrate) and itaconate (inhibitor) in the respective subunits. To identify structural features of tbICL in complex with succinate and itaconate, we compared our structure with that of others, such as the native form of tbICL (PDB code: 1F61) [21] and the complex forms with glyoxylate and succinate (PDB code: 1F8I) [21] and with pyruvate (PDB code: 1F8M) [21]. Structural comparison analysis revealed that the overall structures of the four conformers are similar (RMSD = 0.29–1.47 Å over 400–416 Cα). However, we found that the positions of the L5 loop (residues 189–197) and the C-terminal tail are different, depending on the conformer. Most intriguingly, our structure (subunit A) is nearly identical to the native form

**(A)**

**(B)**

**(C)**

**(D)**

**(E)**

**(F)**

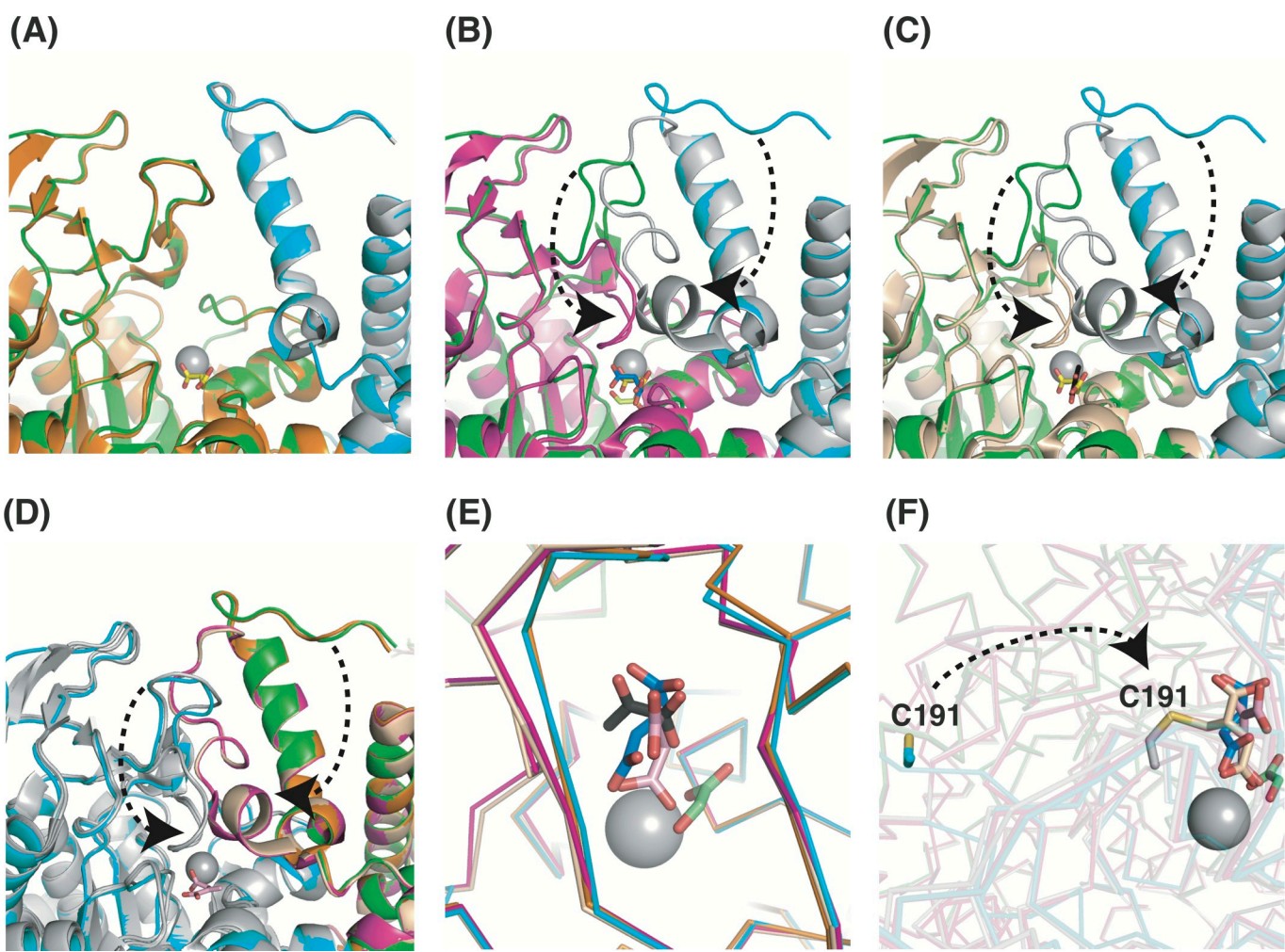

**Fig 3. Structural comparison with other conformers.** Structural comparison of the L5 loop and the C-terminal tail. Subunit A of our tbICL structure (green) is superimposed onto those of the native form (A; orange), glyoxylate and succinate complex (B; magenta), and pyruvate complex (C; wheat) structures. Subunits B are colored gray except for our structure (cyan). All views are restricted to the active site. Succinate from our structure (yellow), glyoxylate (limon), succinate (blue; PDB code: 1F8I), pyruvate (black), and Mg$^{2+}$ ions (gray) are represented by sticks and spheres. The curved arrows indicate conformational changes of the L5 loop and the C-terminal tail. (D) Subunit B of our tbICL structure (cyan) is superimposed onto those of the structures shown in panel (C). Subunit B, except for our structure, is colored gray for clarity. The pink stick represents itaconate. (E) Magnified view of the active site. Stick symbols and color codes are the same as in panels (A)–(D). (F) Magnified view of the active site in subunit B of our structure, the succinate/glyoxylate-complex structure, and another itaconate (wheat) complex structure (PDB code: 6XPP). Stick symbols and color codes are the same as in panels (E).

(PDB code: 1F61; Fig 3A), although our structure contains succinate in the active site. However, comparative analysis with the other complex forms exhibits remarkable structural differences in the positions of the L5 loop and the C-terminal tail. Although the two parts are positioned away from the Mg$^{2+}$ ion at the active site in our structure, they are very close to it in the two complex structures (Fig 3B and 3C). Such structural differences were identical to those found in subunit B, containing itaconate in the active site (Fig 3D).

These findings raise the issue of what causes the conformational changes. As shown in Fig 3E, succinate and itaconate in our structure share nearly the same spatial coordinates as glyoxylate and succinate in the previous complex structure (PDB code: 1F8I) [21]. Moreover, our structure corresponds to the same open form as that observed in the native form, although subunit B has the inhibitor itaconate in the active site. Considering that itaconate is positioned

at nearly the identical site as those of the known compounds, it is unlikely that the conformational changes of the L5 loop and the C-terminal tail depend on the type of inhibitor. It is noteworthy that our dimeric structure has different compounds at the two active sites, whereas the other multimeric complex structures (PDB codes: 1F61, 1F8I, and 1F8M) have the same compounds at their equivalent active sites. This difference implies that the homogeneity of compounds binding to the active site might be a crucial factor for inducing conformational changes.

A recent study showed a tbICL structure in complex with itaconate (PDB code: 6XPP), which is covalently bound to Cys191 [32]. Considering that the chemical state of itaconate in their study is different from that in the current study, structural comparison can provide valuable information on the conformation of itaconate in the active site. Hence, we superimposed the covalently bound itaconate in their structure onto that in our structure. The result showed that the $C_3$, $C_4$, $C_5$, $O_3$, and $O_4$ atoms of itaconate are positioned differently from each other (Fig 3F), i.e., access of Cys191 in the L5 loop to the active site resulted in the formation of a covalent bond with the $C_4$ atom of itaconate. This difference in the stereochemistry of the two itaconates signifies that itaconate can adopt multiple conformations in response to the position of the L5 loop.

## A possible determinant of active site closure

To investigate the biophysical properties of the active site, we analyzed the B-factor profile of tbICL, which can provide information on its intrinsic flexibility. This revealed that the L5 loops in each subunit exhibit high B-factor values, and the C-terminal tails correspond to disordered regions (Fig 4A). We also found that the B-factor profile of the L5 loop in subunit A exhibits slightly higher values than that in subunit B. This difference could result from the different compounds bound to their respective active sites. For example, itaconate bound to the active site in subunit B may have contributed to greater structural stabilization than succinate bound to the active site in subunit A, although neither itaconate nor succinate gave rise to the type of conformational change observed in previous complex structures (PDB code: 1F8I and 1F8M).

As shown in Fig 2B, the active site cavity has a strong, negatively charged surface, whereas some adjacent areas have a positively charged surface. Such uneven charge distributions enabled us to investigate electric field generation around the active site. We found that electric fields are generated across the positively charged regions to the negatively charged active site (Fig 4B). Notably, we found that while the space around the active sites in our structure constituted positively charged regions in a slice of the electric field (Fig 4C), the corresponding space in the previous native and complex structures exhibited negatively charged fields (Fig 4D and 4E).

Based on these results, we assume that the incorporation of heterogeneous compounds into their respective active sites induces subtle movements of nearby residues, thereby generating abnormal electric fields and finally obstructing active site closure. If our assumption is correct, the entry of the same substrates (isocitrate-isocitrate) into the active sites may be a prerequisite for the conformational changes of the L5 loop and the C-terminal tail to close the active site. In addition, if our structure provides a snapshot of the inhibitory mode of itaconate targeting tbICL, itaconate may act as an inhibitor by binding only to one of the two active sites of tbICL, which possibly results in asymmetric distributions of rotameric states of residues related to active site closure in the two subunits. At present, it is difficult to confirm that such a difference in the electric field hampers the conformational changes of the L5 loop and the C-terminal tail. Nevertheless, the hypothesis tested in the present study provides novel and profound insights into conformational changes of the active site architecture in response to binding of the substrate and inhibitor of tbICL.

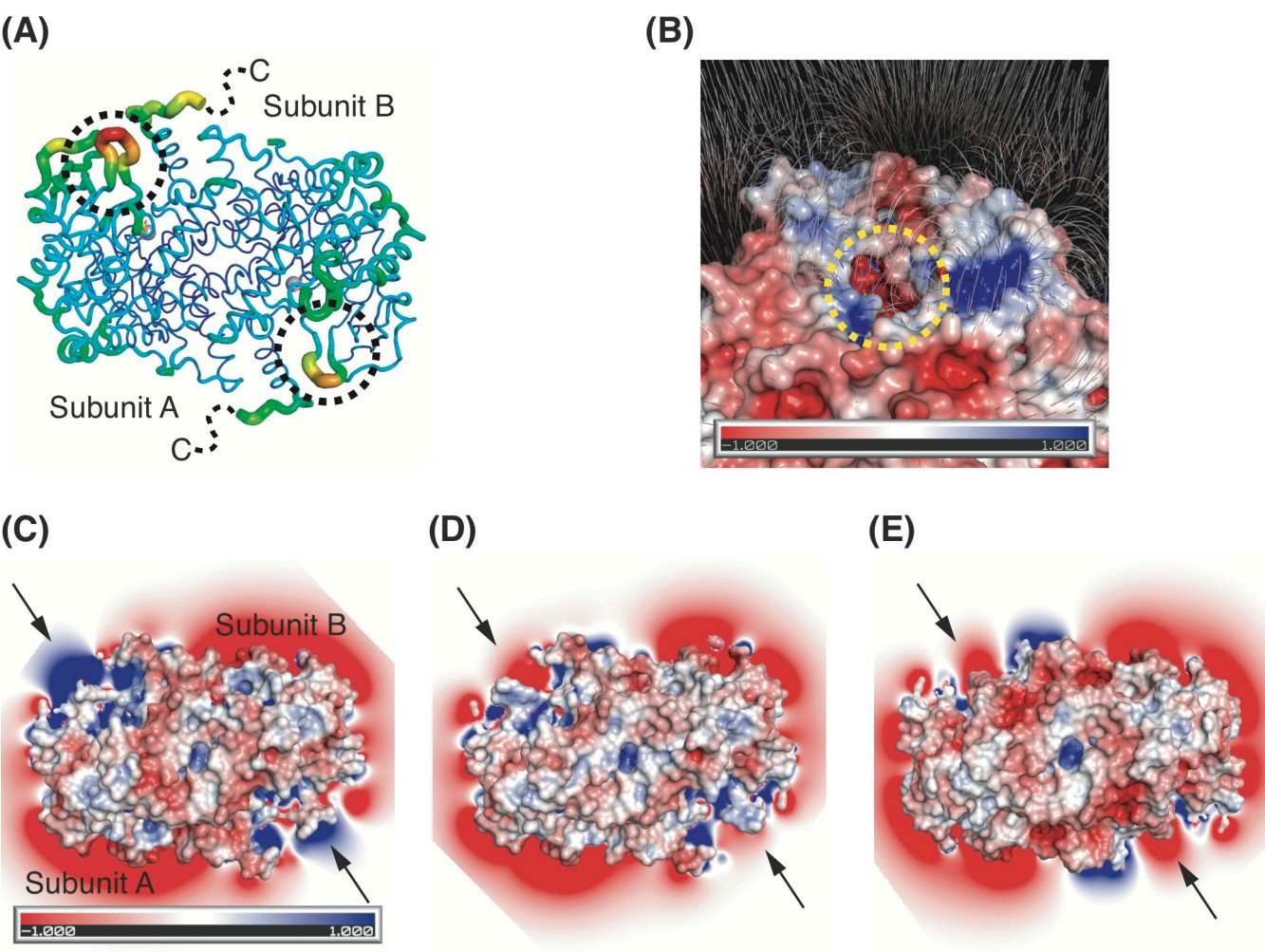

**Fig 4. Biophysical properties around the active site.** (A) B-factor distribution of tbICL in complex with succinate and itaconate. The dimeric structure is depicted in putty representation and colored from red (high) to violet (low) in B-factor value order. The dashed circles indicate the L5 loops. (B) Electric field generation of tbICL in complex with succinate and itaconate. Surface electrostatic potential is represented by the same method as in Fig 2B. The potential gradient map of the electric field is contoured at the 0 sigma level. The scale bar for the electric field ranges from −1 kT/e (red) to 1 kT/e (blue). Electric field distributions of our structure (C), the native structure (D: PDB code: 1F61), and the glyoxylate-succinate complex structure (E; PDB code: 1F8I). The arrows indicate active sites.

## Supporting information

**S1 File. Full wwPDB X-ray structure validation report.**
(PDF)

## Acknowledgments

We thank the staff at 5C beamline of the Pohang Accelerator Laboratory (Pohang, Korea) for their assistance during data collection.

## Author Contributions

**Conceptualization:** Hyun Ho Park.

**Data curation:** Sunghark Kwon, Hye Lin Chun, Hyun Ji Ha, So Yeon Lee.

**Formal analysis:** Sunghark Kwon, Hye Lin Chun.

**Funding acquisition:** Hyun Ho Park.

**Supervision:** Hyun Ho Park.

**Writing – original draft:** Sunghark Kwon, Hye Lin Chun.

**Writing – review & editing:** Hyun Ho Park.

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
