## [Decision Letter · Decision Letter 0]

17 Mar 2021

PONE-D-20-39202

Heterogeneous multimeric structure of isocitrate lyase in complex with succinate and itaconate provides novel insights into its inhibitory mechanism

PLOS ONE

Dear Dr. Park,

Thank you for submitting your manuscript to PLOS ONE. After careful consideration, we feel that it has merit but does not fully meet PLOS ONE’s publication criteria as it currently stands. Therefore, we invite you to submit a revised version of the manuscript that addresses the points raised during the review process.

In your revised version please take into account the criticisms of the two reviewers, especially of the detailed and constructive comments of Reviewer 2. Also, please note the comment of Reviewer 1 concerning the quality of the English. The revised version should be edited by someone with a thorough knowledge of English. 

We look forward to receiving your revised manuscript.

Kind regards,

Israel Silman

Academic Editor

PLOS ONE

Journal Requirements:

2. Please include captions for your Supporting Information files at the end of your manuscript, and update any in-text citations to match accordingly. Please see our Supporting Information guidelines for more information: http://journals.plos.org/plosone/s/supporting-information

Reviewers' comments:

Reviewer's Responses to Questions

**Comments to the Author**

1. Is the manuscript technically sound, and do the data support the conclusions?

Reviewer #1: Yes

Reviewer #2: Partly

2. Has the statistical analysis been performed appropriately and rigorously? 

Reviewer #1: Yes

Reviewer #2: N/A

3. Have the authors made all data underlying the findings in their manuscript fully available?

Reviewer #1: Yes

Reviewer #2: Yes

4. Is the manuscript presented in an intelligible fashion and written in standard English?

Reviewer #1: Yes

Reviewer #2: Yes

5. Review Comments to the Author

Reviewer #1: Authors studied the crystal structure of tbICL in complex with itaconate and succinate. They concluded that that the open conformation of tbICL is due to the binding of both substrate and inhibitor in the active site. The manuscript need to address the recent works by:

1. Ibeji et al. 2020. Demystifying the catalytic pathway of Mycobacterium tuberculosis isocitrate lyase. Sci. Rep. 10: 18925. doi: 10.1038/s41598-020-75799-8

2. Bhusal et al. 2019. Acetyl-CoA-mediated activation of Mycobacterium tuberculosis isocitrate lyase 2. Nat Commun. 10: 4639. doi: 10.1038/s41467-019-12614-7

In addition, grammatical mistakes were noted. The manuscript need to be proof read by professional who is proficient in English.

Reviewer #2: In this manuscript, Kwon and co-workers reported their structural work on understanding the binding of itaconate and succinate to Mtb ICL1. ICLs are important enzymes in Mtb as they regulate the flow of carbon between the glyoxylate shunt and the TCA cycle as well as playing an important role in the methylcitrate cycle for propionate detoxification. The binding of itaconate is in particularly interesting because itaconate is a macrophage metabolite that possesses antimicrobial activity and inhibits ICLs, and it is a succinate analogue that may give us insights into its catalytic mechanism (as ICLs also catalyse the back reaction between succinate, glyoxylate and isocitrate).

From the abstract, I was initially very excited about the manuscript. However, after carefully reading the manuscript, I found the discussions and conclusions of the manuscript too speculative. Key control experiments are missing and important recent discoveries on ICLs were omitted from the discussions. I recommend major revision to the manuscript (which may include the gathering of additional supporting data).

Comments:

(1) Introduction: The Introduction section was relatively well written. However, I have several recommendations. (a) Last sentence in the first paragraph: The involvement of ICLs in the methylcitrate cycle is not universal to all bacteria. Instead, it is limited to certain mycobacterium species. (b) The authors should consider replacing references 7 and 8 (which are research papers covering only a handful of ICL inhibitors) with review articles that highlight the importance of ICLs as a therapeutic target against TB, such as Drug Discov. Today, 2017, 22, 1008-1016, Recent Pat. Inflamm. Allergy Drug Discov. 2013, 7, 114-123. (c) When discussing about ICL inhibitors, the authors could include further reviews including Curr. Med. Chem., 2012, 19, 6126-6137. (d) The authors could include a discussion about the two Mtb ICL isoforms as they are structurally very different but both play important roles in the Mtb carbon catabolism. See Nat. Med., 2005, 11, 638-644 and Nat. Commun., 2019, 10, 4639.

(2) Methods: The ligand-bound crystals were obtained by soaking. However, it is not clear where the succinate comes from. Succinate is not included in the crystallography or purification buffers. Does succinate has a high binding affinity to ICL? It’s interesting to see the presence of succinate in an open-form binding site after rounds of purification and dilution. Also, when was the Mg2+ ion introduced?

(3) Results and discussion: (a) Given the manuscript was submitted in December, I am surprised that the manuscript omitted a ICL-itaconate structure paper that was published in October last year (RSC Med. Chem. 2021, 12, 57-61). In the RSC Med. Chem. paper, the ICL-itaconate structure showed that itaconate is a covalent inhibitor (the structure was obtained through co-crystallisation) and in the current manuscript, itaconate appears to bind non-covalently but the structure was obtained by soaking. Comparison of the two structures is therefore very interesting because it may offer insights into how itaconate may bind to ICLs before the covalent reaction occurs. For example, itaconate appears to be binding very differently when it is bound covalently and non-covalently. More interestingly, as succinate and itaconate are structural analogues, it is interesting that they appear to bind different to the ICL active site. In the RSC Med. Chem. paper, the authors showed that the presence of glyoxylate may speed up covalent reactions - I wonder if itaconate may bind to ICLs with multiple conformations in the absence of glyocylate (and the different binding orientations of itaconate and succinate are showing these possible conformations) - Hence, I would recommend the authors to try soaking with BOTH itaconate/succinate and glyoxylate, as it may help explain the differences between the covalent and non-covalent structures. Also, it is useful to have a superimposed figure comparing the stereochemistry of the two bound itaconates (and succinate) with that in 1F8I. This backs the theory of a properly bound ligand failed to induce active site closure because of the heterogeneous succinate in the adjacent subunit. This would cross off the possibility of having the itaconate merely ‘sitting’ in the binding pocket with succinate in a ‘released’ state. (b) I am not surprised that the ICL1 was found in an open conformation as all previous closed conformation structures were obtained by co-crystallisation (Nat. Struct. Biol., 2000, 7, 663-668 and the RSC Med. Chem. paper) whilst soaking was used in this manuscript.

(4) Results and discussion: We (as in the community) still do not know exactly how the active site loop, the C term tail and the ligand coordinates to close the active site. Based on the structures (with soaking) I think the discussion about transition from the open to close conformations is too speculative. If the authors really want to study the effect, I would recommend conducting co-crystallisation experiments with itaconate/succinate with ICL1 (but with the cysteine mutated to a serine) so that the authors could get a snapshot about the active site loop after itaconate is bound but before the covalent reaction occurs. If the author’s hypothesis is true, that means all four monomers are working in sync. I wonder how it might be in ICL2? I understand that these experiments are probably out of scope of this study but these are needed if the authors want to talk about the closure of the active sites etc.

(5) Overall, I think this manuscript is of potential interest to the community but I think the authors should consider the feedback above (especially in the discussion) when revising the manuscript.

6. PLOS authors have the option to publish the peer review history of their article (what does this mean?). If published, this will include your full peer review and any attached files.

Reviewer #1: No

Reviewer #2: No

---

## [Author Response · Author response to Decision Letter 0]

11 Apr 2021

Manuscript No.: PONE-D-20-39202

"Heterogeneous multimeric structure of isocitrate lyase in complex with succinate and itaconate provides novel insights into its inhibitory mechanism"

By Kwon et al.

Our revision and reply to the comments

Reviewer #1’s comment:

Authors studied the crystal structure of tbICL in complex with itaconate and succinate. They concluded that that the open conformation of tbICL is due to the binding of both substrate and inhibitor in the active site. The manuscript need to address the recent works by:

1. Ibeji et al. 2020. Demystifying the catalytic pathway of Mycobacterium tuberculosis isocitrate lyase. Sci. Rep. 10: 18925. doi: 10.1038/s41598-020-75799-8

2. Bhusal et al. 2019. Acetyl-CoA-mediated activation of Mycobacterium tuberculosis isocitrate lyase 2. Nat Commun. 10: 4639. doi: 10.1038/s41467-019-12614-7

In addition, grammatical mistakes were noted. The manuscript need to be proof read by professional who is proficient in English.

Our revision and reply:

As suggested by the reviewer, we have addressed the two recent works in Introduction (page 3, lines 36-38, 40-43 in the revised manuscript). In addition, the manuscript has been proofread by a professional editor who is proficient in English. 

Reviewer #2’s comment (1):

(1) Introduction: The Introduction section was relatively well written. However, I have several recommendations. (a) Last sentence in the first paragraph: The involvement of ICLs in the methylcitrate cycle is not universal to all bacteria. Instead, it is limited to certain mycobacterium species. (b) The authors should consider replacing references 7 and 8 (which are research papers covering only a handful of ICL inhibitors) with review articles that highlight the importance of ICLs as a therapeutic target against TB, such as Drug Discov. Today, 2017, 22, 1008-1016, Recent Pat. Inflamm. Allergy Drug Discov. 2013, 7, 114-123. (c) When discussing about ICL inhibitors, the authors could include further reviews including Curr. Med. Chem., 2012, 19, 6126-6137. (d) The authors could include a discussion about the two Mtb ICL isoforms as they are structurally very different but both play important roles in the Mtb carbon catabolism. See Nat. Med., 2005, 11, 638-644 and Nat. Commun., 2019, 10, 4639.

Our revision and reply:

(a) We have rectified the corresponding sentence as follows: “ICLs from Mycobacterium tuberculosis can also catalyze the lysis of 2-methylisocitrate, to produce pyruvate and succinate” (page 3, lines 9-11 in the revised manuscript).

(b) We have replaced the corresponding references (7 and 8) with the two review papers recommended by the reviewer (page 3, lines 17 in the revised manuscript).

(c) We have added the review paper suggested by the reviewer to the manuscript (page 3, lines 19 in the revised manuscript).

(d) As suggested by the reviewer, we have included the discussion on tbICL isoforms in the manuscript (page 3, lines 36-38 in the revised manuscript).

Reviewer #2’s comment (2):

(2) Methods: The ligand-bound crystals were obtained by soaking. However, it is not clear where the succinate comes from. Succinate is not included in the crystallography or purification buffers. Does succinate has a high binding affinity to ICL? It’s interesting to see the presence of succinate in an open-form binding site after rounds of purification and dilution. Also, when was the Mg2+ ion introduced?

Our revision and reply:

We inadvertently omitted information on where the succinate came from in the previous manuscript. For soaking, crystallization buffers were supplemented with respective compounds (5 mM) in various combinations: 1) itaconate, 2) isocitrate, 3) succinate, 4) glyoxylate, 5) succinate + glyoxylate, 6) isocitrate + itaconate, 7) succinate + itaconate, 8) glyoxylate + itaconate, and 8) succinate + glyoxylate + itaconate. Our structure containing succinate and itaconate was obtained from the 7th combination. Most crystals with the other combinations showed relatively poor resolution or ambiguous map density in the active site. In addition, we assume that the Mg2+ ion was probably originated from the LB medium, because we did not supply it intentionally in purification, crystallization, and soaking steps. We added this content to the manuscript (page 5, lines 3-7; page 5, lines 32; page 6, lines 12-13; page 6, lines 22-24 in the revised manuscript).

Reviewer #2’s comment (3):

(3) Results and discussion: (a) Given the manuscript was submitted in December, I am surprised that the manuscript omitted a ICL-itaconate structure paper that was published in October last year (RSC Med. Chem. 2021, 12, 57-61). In the RSC Med. Chem. paper, the ICL-itaconate structure showed that itaconate is a covalent inhibitor (the structure was obtained through co-crystallisation) and in the current manuscript, itaconate appears to bind non-covalently but the structure was obtained by soaking. Comparison of the two structures is therefore very interesting because it may offer insights into how itaconate may bind to ICLs before the covalent reaction occurs. For example, itaconate appears to be binding very differently when it is bound covalently and non-covalently. More interestingly, as succinate and itaconate are structural analogues, it is interesting that they appear to bind different to the ICL active site. In the RSC Med. Chem. paper, the authors showed that the presence of glyoxylate may speed up covalent reactions - I wonder if itaconate may bind to ICLs with multiple conformations in the absence of glyocylate (and the different binding orientations of itaconate and succinate are showing these possible conformations) - Hence, I would recommend the authors to try soaking with BOTH itaconate/succinate and glyoxylate, as it may help explain the differences between the covalent and non-covalent structures. Also, it is useful to have a superimposed figure comparing the stereochemistry of the two bound itaconates (and succinate) with that in 1F8I. This backs the theory of a properly bound ligand failed to induce active site closure because of the heterogeneous succinate in the adjacent subunit. This would cross off the possibility of having the itaconate merely ‘sitting’ in the binding pocket with succinate in a ‘released’ state. (b) I am not surprised that the ICL1 was found in an open conformation as all previous closed conformation structures were obtained by co-crystallisation (Nat. Struct. Biol., 2000, 7, 663-668 and the RSC Med. Chem. paper) whilst soaking was used in this manuscript.

Our revision and reply:

(a) As mentioned in our revision and reply to the reviewer #2’s comment (2), we tried various soaking experiments but obtained crystals with diffraction-quality in the combination of succinate and itaconate. As further work, we plan to explore other crystallization conditions in the presence of glyoxylate. We also think that the recent paper (RSC Med. Chem. 2021, 12, 57-61) can provide valuable information on the conformation of itaconate in the active site. As recommended by the reviewer, we have superimposed the covalently bound itaconate onto that in our structure. This result showed that the C3, C4, C5, O3, and O4 atoms of itaconate are positioned differently from each other. Namely, access of Cys191 in the L5 loop to the active site resulted in the formation of a covalent bond with the C4 atom of itaconate. This difference of the stereochemistry of the two itaconates signifies that itaconate can adopt multiple conformations in response to the position of the L5 loop. We added the superimposition figure and this discussion to Figure 3F and the manuscript, respectively (page 8, lines 35-45 in the revised manuscript).

(b) The open form observed in our structure is assumed to result from the binding of the disparate compounds to the respective active sites.

Reviewer #2’s comment (4):

(4) Results and discussion: We (as in the community) still do not know exactly how the active site loop, the C term tail and the ligand coordinates to close the active site. Based on the structures (with soaking) I think the discussion about transition from the open to close conformations is too speculative. If the authors really want to study the effect, I would recommend conducting co-crystallisation experiments with itaconate/succinate with ICL1 (but with the cysteine mutated to a serine) so that the authors could get a snapshot about the active site loop after itaconate is bound but before the covalent reaction occurs. If the author’s hypothesis is true, that means all four monomers are working in sync. I wonder how it might be in ICL2? I understand that these experiments are probably out of scope of this study but these are needed if the authors want to talk about the closure of the active sites etc.

Our revision and reply:

We appreciate the reviewer’s helpful advice and suggestions for further work. As of now, however, it seems somewhat difficult to simultaneously conduct the co-crystallization experiments, accompanying mutagenesis, with ICL2 as well as ICL1. We would sincerely plan to perform what the reviewer recommended as our further work.

Reviewer #2’s comment (5):

(5) Overall, I think this manuscript is of potential interest to the community but I think the authors should consider the feedback above (especially in the discussion) when revising the manuscript.

Our revision and reply:

We appreciate the reviewer’s evaluation of our manuscript.

---

## [Decision Letter · Decision Letter 1]

20 Apr 2021

Heterogeneous multimeric structure of isocitrate lyase in complex with succinate and itaconate provides novel insights into its inhibitory mechanism

PONE-D-20-39202R1

Dear Dr. Park,

We’re pleased to inform you that your manuscript has been judged scientifically suitable for publication and will be formally accepted for publication once it meets all outstanding technical requirements.

Kind regards,

Israel Silman

Academic Editor

PLOS ONE

Additional Editor Comments (optional):

Reviewers' comments:

Reviewer's Responses to Questions

**Comments to the Author**

1. If the authors have adequately addressed your comments raised in a previous round of review and you feel that this manuscript is now acceptable for publication, you may indicate that here to bypass the “Comments to the Author” section, enter your conflict of interest statement in the “Confidential to Editor” section, and submit your "Accept" recommendation.

Reviewer #2: All comments have been addressed

2. Is the manuscript technically sound, and do the data support the conclusions?

Reviewer #2: Yes

3. Has the statistical analysis been performed appropriately and rigorously? 

Reviewer #2: N/A

4. Have the authors made all data underlying the findings in their manuscript fully available?

Reviewer #2: Yes

5. Is the manuscript presented in an intelligible fashion and written in standard English?

Reviewer #2: Yes

6. Review Comments to the Author

Reviewer #2: I am satisfied with the changes that the authors made to their manuscript. Happy to recommend publication with minor changes listed below:

(1) The structures of isocitrate and succinate shown in Figure 1(a) are not correct

(2) Minor English checks needed. For example, in the abstract, “During the glyoxylate cycle” should be “In the glyoxylate cycle”

(3) I recommend better labelling of the structure figures (esp 3 and 4) to highlight the active site, C-terminal loop, key amino acids at the active site and so on

7. PLOS authors have the option to publish the peer review history of their article (what does this mean?). If published, this will include your full peer review and any attached files.

Reviewer #2: **Yes: **Ivanhoe Leung

---

## [Editor Report · Acceptance letter]

26 Apr 2021

PONE-D-20-39202R1 

Heterogeneous multimeric structure of isocitrate lyase in complex with succinate and itaconate provides novel insights into its inhibitory mechanism 

Dear Dr. Park:

I'm pleased to inform you that your manuscript has been deemed suitable for publication in PLOS ONE. Congratulations! Your manuscript is now with our production department. 

Kind regards, 

on behalf of

Prof. Israel Silman 

Academic Editor

PLOS ONE